# A 12 kb multi-allelic copy number variation encompassing a GC gene enhancer is associated with mastitis resistance in dairy cattle

**Young-Lim Lee**[1]*, **Haruko Takeda**[2], **Gabriel Costa Monteiro Moreira**[2], **Latifa Karim**[3], **Erik Mullaart**[4], **Wouter Coppieters**[2,3], **The GplusE consortium**[5], **Ruth Appeltant**[2], **Roel F. Veerkamp**[1], **Martien A. M. Groenen**[1], **Michel Georges**[2], **Mirte Bosse**[1], **Tom Druet**[2], **Aniek C. Bouwman**[1], **Carole Charlier**[2]

1 Wageningen University & Research, Animal Breeding and Genomics, Wageningen, the Netherlands,
2 Unit of Animal Genomics, GIGA-R & Faculty of Veterinary Medicine, University of Liège, Liège, Belgium,
3 GIGA Genomics Platform, GIGA Institute, University of Liège, Liège, Belgium, 4 CRV B.V., Arnhem, the Netherlands, 5 http://www.gpluse.eu

* younglim.lee@wur.nl

**Data Availability Statement:** Genome sequence data of the CM resistance QTL region (BTA 6:84-93 Mb) of the 266 Dutch Holstein Friesian animals are

## Abstract

Clinical mastitis (CM) is an inflammatory disease occurring in the mammary glands of lactating cows. CM is under genetic control, and a prominent CM resistance QTL located on chromosome 6 was reported in various dairy cattle breeds. Nevertheless, the biological mechanism underpinning this QTL has been lacking. Herein, we mapped, fine-mapped, and discovered the putative causal variant underlying this CM resistance QTL in the Dutch dairy cattle population. We identified a ~12 kb multi-allelic copy number variant (CNV), that is in perfect linkage disequilibrium with a lead SNP, as a promising candidate variant. By implementing a fine-mapping and through expression QTL mapping, we showed that the group-specific component gene (*GC*), a gene encoding a vitamin D binding protein, is an excellent candidate causal gene for the QTL. The multiplicated alleles are associated with increased *GC* expression and low CM resistance. Ample evidence from functional genomics data supports the presence of an enhancer within this CNV, which would exert *cis*-regulatory effect on *GC*. We observed that strong positive selection swept the region near the CNV, and haplotypes associated with the multiplicated allele were strongly selected for. Moreover, the multiplicated allele showed pleiotropic effects for increased milk yield and reduced fertility, hinting that a shared underlying biology for these effects may revolve around the vitamin D pathway. These findings together suggest a putative causal variant of a CM resistance QTL, where a *cis*-regulatory element located within a CNV can alter gene expression and affect multiple economically important traits.

## Author summary

Clinical mastitis (CM) is an inflammatory disease that negatively influences dairy production and compromises animal welfare. Although one major genetic locus for CM

---

deposited in the European Nucleotide Archive under accession PRJEB45439/ERP129554. The RNA-seq data is deposited under EBI ArrayExpress accession E-MTAB-9348 and 9871. The genotype data used for eQTL mapping is available in the supporting information. The ATAC-seq data is deposited under EBI ArrayExpress accession E-MTAB-9872. Genotype and phenotype data of 4,142 bulls used for QTL mapping on BTA 6 are available upon request directed to CRV B.V. (chris.schrooten@crv4all.com) and require a material transfer agreement.

**Funding:** YLL, ACB, MB, MAMG, and RFV are financially supported by the Dutch Ministry of Economic Affairs (TKI Agri & Food project 16022) and the Breed4Food partners Cobb Europe, CRV, Hendrix Genetics and Topigs Norsvin. This work was supported by grants from the European Research Council (Damona to MG; award number: ERC AdG-GA323030), and the EU Framework 7 program (GplusE to MG and HT; award number: 613689). CC is work package leader and GCMM is post-doctoral fellow of the H2020 EU project BovReg (Grant agreement number: 815668). Computational resources have been provided by the Consortium des Équipements de Calcul Intensif (CÉCI), funded by the Fonds de la Recherche Scientifique de Belgique (F.R.S.-FNRS) under Grant No. 2.5020.11 and by the Walloon Region. The funders did not play any role in the study design, data collection and analysis, decision to publish, or preparation of the manuscript.

**Competing interests:** The authors have declared that no competing interest exist.

resistance was mapped on bovine chromosome 6, a mechanistic description of this association has been lacking. Herein, we report a 12-kb multiallelic copy number variant (CNV), encompassing a strong enhancer for group-specific component gene (*GC*), as a likely causal variant for this locus. This CNV is associated with high *GC* expression and low CM resistance. We speculate that upregulation of *GC* leads to a large amount of vitamin D binding protein, which in turn, reduces biologically available vitamin D, leading to low CM resistance. Despite the negative effect on CM resistance, the CNV contributes to increased milk production, hinting at balancing selection. Our results highlight how multiplication of a regulatory element can shape economically important traits in dairy cattle, both in favourable and unfavourable directions.

## Introduction

Clinical mastitis (CM) is an inflammation in the mammary glands. This condition is often seen in dairy cattle and the repercussions of CM include production loss, use of antibiotics, and compromised animal welfare [1]. CM resistance has a genetic component, with estimated heritabilities ranging between 0.01 and 0.10 [2–6]. Genome-wide associations studies (GWAS) identified several quantitative trait loci (QTL) associated with CM resistance or somatic cell score (SCS), an indicator trait of CM [2–7]. For instance, the six most significant CM resistance QTLs together capture 8.9% of the genetic variance in Danish Holstein Friesian (HF) cattle, underlining the polygenic nature of CM resistance [7]. Of these six QTL, the most significant QTL has been mapped near 88 Mb on the Bos taurus autosome (BTA) 6 in various dairy cattle populations [8–11], including Dutch HF [12]. Several fine-mapping studies, using imputed whole genome sequence (WGS) variants, reported non-coding candidate causal SNPs at the group-specific component (*GC*) gene [7,11,13–15]. One of these studies investigated, albeit unsuccessfully, whether one of the non-coding candidate SNP obtained from the GWAS was associated with *GC* expression, leaving the functional mechanisms underlying this association elusive [13]. Interestingly, the proposed candidate SNPs showed antagonistic allele effects for milk yield (MY) (the high CM resistance allele was linked to low MY, and vice versa [7,13,16]), and one study concluded that a single pleiotropic variant regulates both of the traits [17]. Furthermore, this locus harbours QTL for many traits including body conformation, fertility, and longevity [11,15,18–22], implying pleiotropy, which remains to be investigated. Until now, *GC*, a gene that encodes the vitamin D binding protein (DBP), has been considered the most promising candidate gene for the CM resistance QTL on BTA 6 [13,14]. A growing body of literature underpins the importance of DBP, which acts as a macrophage activating factor, modulates immune responses [23], and is central to the vitamin D pathway [24]. For instance, polymorphisms in *GC* have been shown to cause vitamin D deficiency and inflammatory diseases in humans [25]. Moreover, a therapeutic use of vitamin D in lactating, CM-infected cows reduced inflammation, implying a link between vitamin D and inflammation [26–28]. Yet, the GWAS lead SNP was not associated with *GC* expression or alternative transcription, leaving the functional mechanism elusive [13]. Some researchers hypothesized that a copy number variant (CNV) might be the causal variant underlying this QTL [7]. Indeed, a CNV in high linkage disequilibrium (LD) with the GWAS lead SNP was found in the 3' alternative exon of *GC*, however, the functional role of this CNV was not well characterized [13]. In this study, we aimed at fine-mapping the prominent CM resistance QTL in the Dutch HF population, and identifying a candidate causal gene and a variant that may explain the functional mechanism(s) of the QTL. Our findings show that (1) the CM resistance QTL on BTA 6 is

confirmed in our Dutch HF population; (2) a 12-kb multi-allelic CNV, encompassing the 3' alternative exon of *GC*, harbours a putative enhancer, which exerts *cis*-regulation on the candidate causal gene, *GC*; (3) a haplotype associated with the CNV allele is strongly selected for, and the CNV has pleiotropic effects on MY, body conformation and fertility traits. These findings together highlight a functional CNV that contains a *cis*-regulatory element, affecting gene expression and subsequently altering the economically important traits.

## Results

### A major CM resistance QTL on BTA 6 segregates in the Dutch HF cattle population

A strong CM resistance QTL has been identified on BTA6 in several cattle populations, including Dutch HF [7–14]. To fine-map this QTL in the Dutch HF population, we first performed an association analysis on BTA6 using 4,142 progeny tested bulls. These animals were genotyped using a custom 16K array, and imputed sequentially, firstly to the Illumina Bovine 50K array, and then to higher density (770K). All analyses were performed according to the Bovine genome assembly UMD3.1 [29] (See S1 Table for SNP positions in ARS-UCD1.2 genome). De-regressed estimated breeding values of CM resistance were used as phenotypes in a single SNP association analysis (materials and methods). As expected, we replicated the strong association signal near BTA 6:88.6 Mb found in Norwegian Red [13], Danish HF [7,14], and Dutch HF populations [12] (-$\log_{10}$P = 7.35; Fig 1A, S2 Table). This association signal was

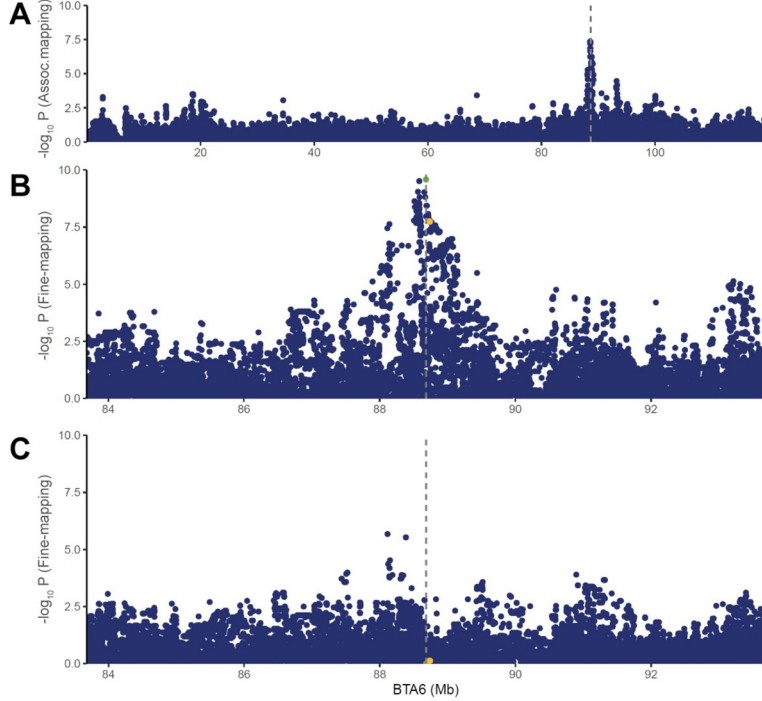

**Fig 1. Association mapping and fine-mapping of the major clinical mastitis resistance QTL on BTA6.** (A) Association mapping performed with imputed BovineHD variants on BTA6. The association signal near BTA 6:88.6 Mb, shown in other dairy cattle populations was replicated in the current Dutch HF population. (B) Fine-mapping performed with imputed WGS variants in BTA 6:84–93 Mb region. A strong association signal was shown in a 200-Kb region (BTA 6:88.5–88.7 Mb), spanning over *GC* gene. Our lead SNP (rs110813063, marked with green and vertical dotted line) has not been reported as a candidate SNP in other CM fine-mapping studies. CM candidate SNPs from other fine-mapping studies are marked as yellow. (C) Conditional analyses including GC CNV as a covariate nullify the association signal.

found downstream of *GC*, a reverse-oriented gene located at BTA 6:88.68–88.74 Mb. We then focused on a 10-Mb window encompassing the QTL (BTA 6:84–93 Mb), using 45,782 imputed WGS level variants present in the window, aiming at (1) confirming the *GC* as a positional candidate gene and (2) identifying a candidate causal variant. This 10-Mb window contained the previously reported CNV, which was in high LD with the GWAS lead SNP [13], and included SNPs located within the CNV in the WGS imputation panel, as they could possibly tag the CNV. The association signal peaked in a 200-Kb region (BTA 6:88.5–88.7 Mb), spanning over *GC* and the region downstream of *GC*. The lead SNP, rs110813063, located at BTA 6:88,683,517 ($-\log_{10}P = 9.59$) was ~4 kb downstream of *GC* (Fig 1B). The T allele of the lead SNP (allele frequency (AF) = 0.58), was associated with low CM resistance, whereas the C allele was associated with high CM resistance (AF = 0.42). Finally, a conditional analysis was performed by including the lead SNP as a covariate in the association model. SNPs in the association peak were no longer significant, with the exception of a minor signal at the left side of the association peak (Fig 1C). Our results confirmed the presence of the CM resistance QTL in the Dutch HF population, however, as with previous studies [7,13–15], the non-coding lead SNP (rs110813063) did not have any evidence of a functional role. There were no coding variants amongst the variant in high LD with the lead SNP ($r^2 > 0.9$). Provided that the significant association signals do not necessarily indicate causality, [30], we further characterized the lead SNP using WGS data.

## A multi-allelic CNV is in high LD with the lead associated SNP for CM resistance QTL on BTA 6

Our lead associated SNP (rs110813063, hereafter shorten as lead SNP) was located within a ~12 kb CNV encompassing the 14th exon of *GC* (Fig 2A). This CNV, present in both dairy and

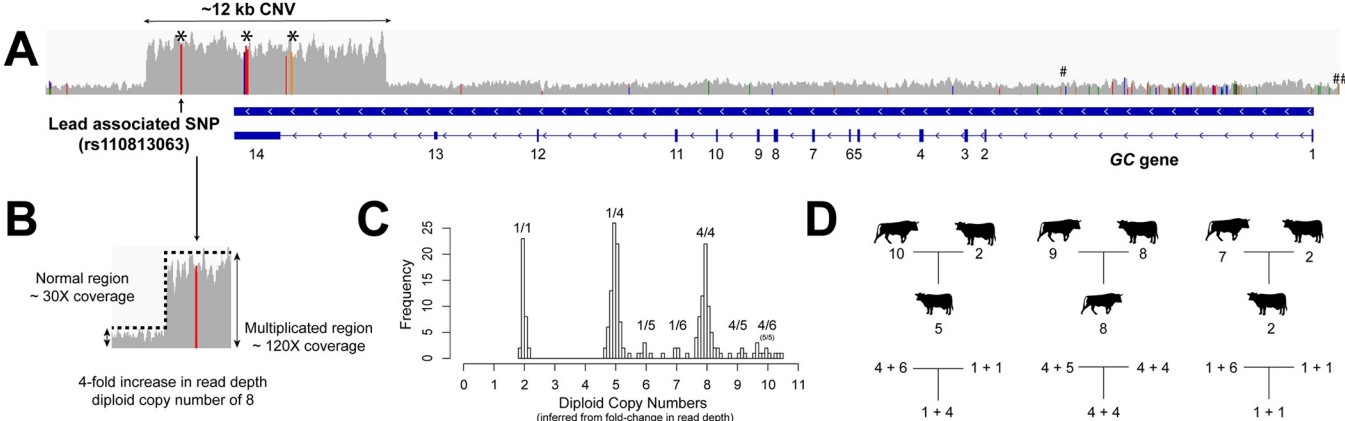

**Fig 2. Discovery of multiallelic GC CNV using deeply sequenced genomes and familial structure.** (A) Schematic overview showing the lead associated SNP and ~12 kb CNV overlapping with *GC*. *GC* is a reverse oriented gene, consisting of 14 exons, of which two last exons are non-coding. Five CNV tagging SNPs were present within the GC CNV and marked with black asterisks (the middle asterisk covers three tagging SNPs). Among them, the first SNP, which was also the lead SNP, was in perfect LD with the GC CNV ($r^2 = 1$), whereas the rest were in high LD ($r^2 > 0.98$). The hash marks at the upstream and the intronic region of *GC* indicate CM resistance candidate SNPs reported by others [13,14]. (B) Sequencing depth difference between the CNV region and normal region was used to infer copy numbers. (C) A histogram of fold change in read depth values shows that majority of animals fall into diploid copy number of 2, 5 and 8, and some minor peaks occur at diploid copy number of 6, 7, 9 and 10. Based on this diploid CNs, we inferred haploid CNs of 1, 4, 5, and 6. We showed possible allelic combination(s) above each diploid CN. The diploid CN10 could be comprised of either CN5/CN5 or CN4/CN6; however, our results showed that it was always CN4/CN6. (D) Familial information and background haplotypes were used to phase the copy number and thus revealed how the CNV segregates in trios. The upper family tree shown with animal signs stands for diploid copy numbers, and the lower tree shows haploid copy numbers (the phase results of the diploid CNs).

beef cattle populations [31], was reported to be in high LD with the candidate SNP for CM resistance QTL in a Norwegian Red population [13]. Thus, we hypothesized that the CNV might be the causal variant underlying this QTL. Using our CNV calling pipeline, we characterized the CNV using WGS data of 266 HF animals, exploiting split-read, pair-end mapping, and read depth evidence, confirmed the presence of the ~12 kb CNV at BTA 6:88,681,767–88,693,545 (hereafter referred to as GC CNV; S1 Fig). The 14th exon of *GC*, encompassed by the CNV, is a 3' alternative exon, only accounting for minority of the total *GC* expression [13]. The distribution of normalized read-depth of the CNV region suggested a multi-allelic locus, where putative copy number (CN) alleles include CNs 1, 4, 5, and 6 (Fig 2B and 2C). We determined corresponding CN genotypes (e.g., individuals with CN 2 and 6 would be respectively carriers of alleles CN1/CN1 and CN1/CN5) for all but those individuals with 10 copies, that could be either CN4/CN6 or CN5/CN5. In all sequenced duos or trios, these inferred genotypes were compatible with Mendelian segregation rules (Fig 2D), and no genotype incompatibility was observed. Genotypes from relatives of individuals with 10 copies allowed us also to deduce that all carriers of 10 copies were CN4/CN6. Carriers of alleles CN5 or CN6 were restricted to a limited number of families and the segregation of these two alleles perfectly matched haplotype transmission from parents to offspring (S2 Fig). An analysis of homozygosity-by-descent (HBD) in all sequenced individuals revealed that alleles CN 4, 5 and 6 share a common haplotype, identical-by-descent for at least 200 kb ($\geq$600 SNPs; S3 Table). The HBD segments were rare and extremely short at the CNV position in CN 1 individuals, indicating that this allele was associated with different haplotypes. In summary, we identified four alleles at the GC CNV locus: CN 1 corresponds to a single copy and considered wildtype (Wt) given the high haplotypic diversity, whereas alleles CNs 4–6 correspond to multiple copies (Mul), with four to six copies (Fig 3A). In our population, CN 1 and CN 4 were the most frequent alleles (0.39 and 0.54, respectively), while CN 5 and CN 6 were rare (0.03 and 0.05, respectively; Fig 3A). Furthermore, the GC CNV, coded as a biallelic variant, where CN 1 (Wt) was the reference allele and CNs 4–6 (Mul) were grouped together as the alternative allele, was in perfect linkage ($r^2 = 1$) with the lead SNP (rs110813063). The C allele, associated with high CM resistance, tagged CN1, whereas the T allele, associated with low CM resistance, tagged CNs 4–6. Thus, this lead SNP was used as a surrogate marker for the GC CNV in subsequent analyses. The GC CNV contained five tagging SNPs, including the lead SNP ($r^2 \geq 0.98$; Fig 3B). These five tagging SNPs showed an allelic imbalance pattern in WGS data of Wt/Mul individuals (Fig 3C) due to a disproportionally high number of reads supporting alternative alleles on the multiplication haplotypes. There was no SNP uniquely tagging CNs 4–6 separately, due to their similar and recent haplotypic background.

## Recent positive selection strongly favoured a haplotype harbouring the GC CNV

In livestock breeding, artificial selection for economically important traits (i.e. high CM resistance) potentially drives desired alleles to fixation and removes alleles with negative effects (i.e. low CM resistance) from the population. Thus, it is intriguing that the allele associated with low CM resistance (CNs 4–6) is highly frequent in Dutch HF cattle (combined AF = 0.58). We postulated two alternative hypotheses: (1) GC CNV is pleiotropic, conferring positive effects on different traits under selection, contrary to a negative effect on CM resistance, or (2) GC CNV is in high LD with a causal variant of a strongly selected trait, and therefore, genetic hitch-hiking increased the frequency of the low CM resistance allele. In both cases, the GC CNV would be associated with a selected haplotype. Thus, we scanned BTA 6 for selection signatures based on integrated Haplotype Score (iHS [32,33]), using WGS haplotypes from the

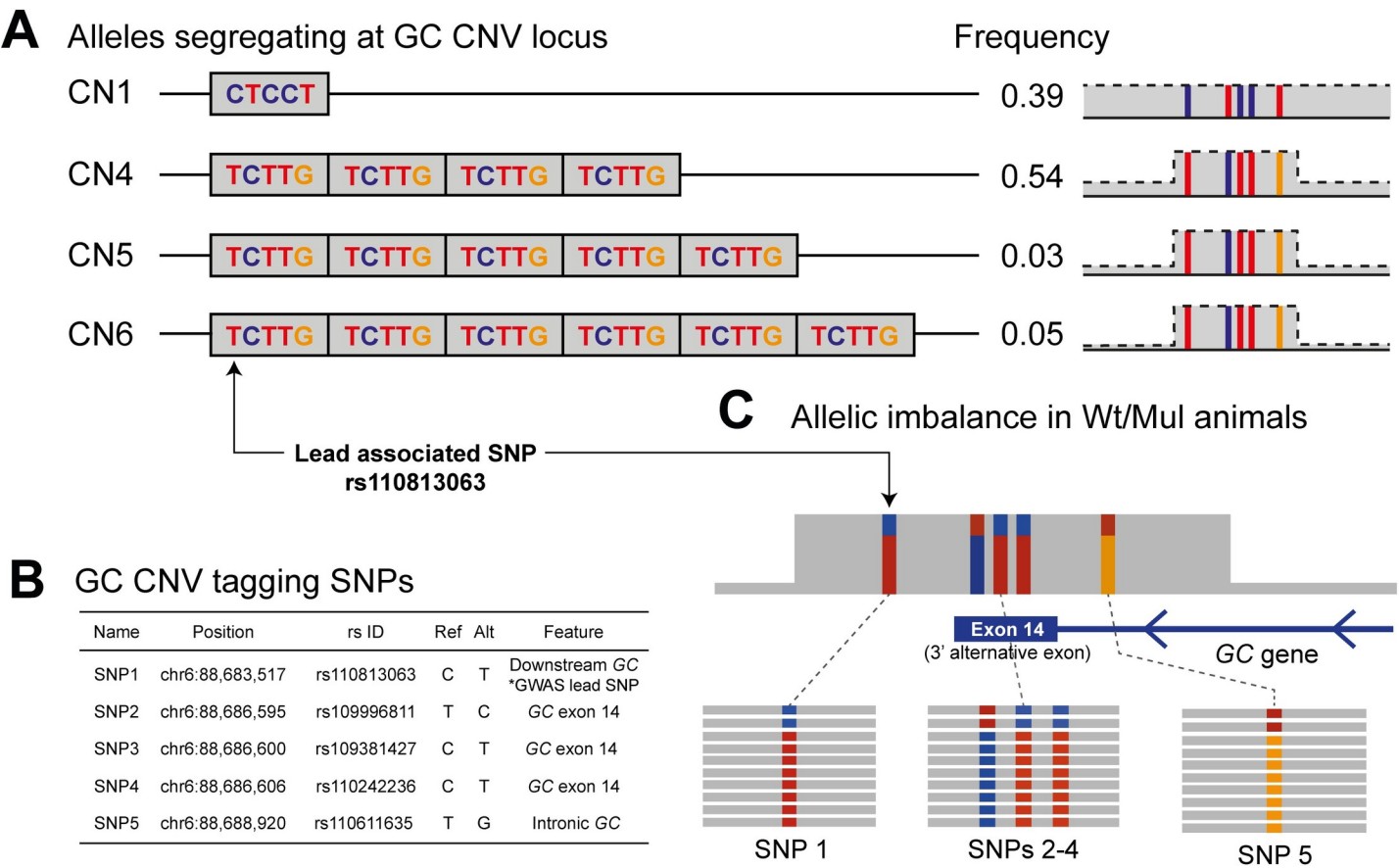

**Fig 3. Characterization of the GC CNV tagging SNPs and allelic imbalance pattern.** (A) A schematic overview of four structural haplotypes and the five tagging SNPs inside the GC CNV, shown together with allele frequencies. (B) Five GC CNV tagging SNPs, shown with their positions, rs ID, alleles, location within GC. (C) Allelic imbalance pattern shown in Wt/Mul animals. Animals will get more supporting reads for alternative alleles for the five CNV tagging SNPs, thus the tagging SNPs will be called as heterozygous but with strong allelic imbalance.

266 HF animals. Of the two strong signals of selection identified near the 79 and 89 Mb regions ($-\log_{10}P>5$; S3 Fig), the latter was only ~200 kb away from the GC CNV, and thus was further inspected (Fig 4A). The extended haplotype homozygosity (EHH), centred at the iHS lead SNP (BTA 6:88,861,709, AF = 0.28) revealed a strongly favoured haplotype, which extended outwards further than the non-selected haplotypes (Fig 4B). As expected, CNs 4–6 were located in the strongly selected haplotype, whereas CN 1 was in the non-selected haplotypes. In addition, this finding was in line with our HBD results, where homozygous CNV carriers (CNs 4–6) shared long HBD haplotypes, whereas homozygous non-CNV carriers (CN 1) did not. These findings supported our hypothesis that a strong positive selection acted upon the region containing the GC CNV. Given the antagonistic effects of the CM resistance QTL on MY [13] and relevance to dairy cattle breeding [34], we deemed MY as a potential target of the selection signature we identified. To confirm whether CM resistance and MY are modulated by the same variant (pleiotropy) or two different variants (LD), the 10-Mb window (BTA 6:84–93 Mb) harbouring the GC CNV was fine-mapped for MY. A strong association signal appeared in 89.08 Mb region, ~400 kb away from the GC CNV ($-\log_{10}P = 7.5$; Fig 4A). The MY lead SNP was found at 89,077,838-bp and the G allele was associated with high MY, whereas the T allele was associated with low MY (AF = 0.45). Regardless of the ~400-Kb

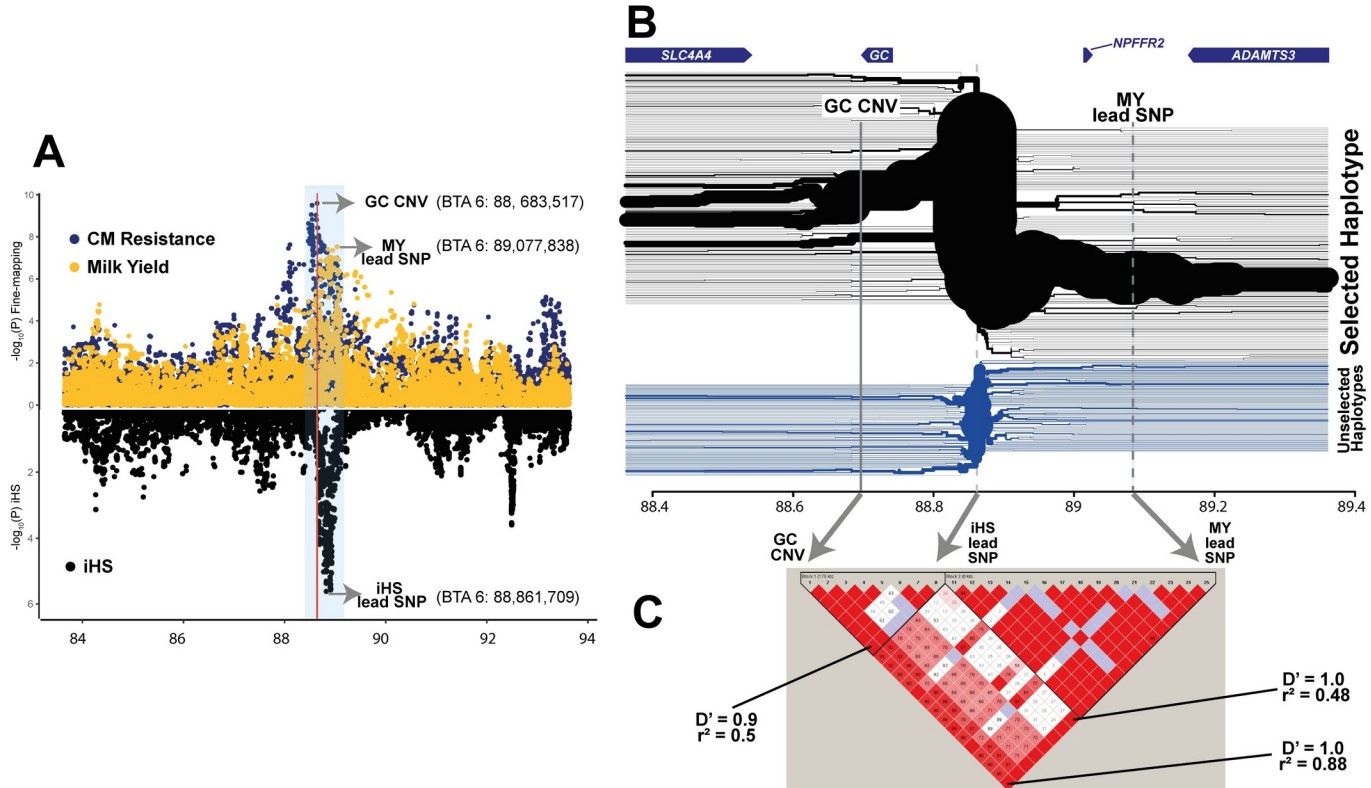

**Fig 4. Selection signature scan and trait association (clinical mastitis resistance and milk yield) plot.** (A) A 10-Mb region with a strong selective signature signal was zoomed in (BTA 6: 84–93 Mb). Association mapping results from imputed WGS variants on CM resistance (dark blue) and MY (yellow) are shown in the upper panel; iHS results are shown in the lower panel (black). The CM resistance association peak occurs at the left side of the iHS peak, whereas MY association peak appears on the right side of the iHS peak. The red vertical line marks GC CNV. A 1-Mb region covering GC CNV, iHS lead SNP, and MY lead SNP are marked with translucent blue. (B) The extended haplotype homozygosity of the 1-Mb region marked in panel (A) is shown, together with four genes annotated in this region (top of the figure). The major haplotype shown in the upper part (black) branches outwards, implying recent positive selection acted upon this haplotype. The non-selected haplotypes, shown in the lower side (blue) rapidly break down from the iHS lead SNP. (C) Pairwise D' and $r^2$ values between GC CNV, iHS lead SNP, and MY lead SNP in the ~4,000 daughter proven bulls. The panel was made using Haploview software [111].

distance, the MY lead SNP and the GC CNV were in high LD ($r^2$ = 0.88). Of note, the iHS lead SNP was located between the association signals for CM resistance and MY (Fig 4A). We re-evaluated the EHH results and found that the strongly selected haplotype harbours CNs 4–6 alleles of the GC CNV and the G allele of the MY lead SNP, implying that the strong selection resulted in low CM resistance and high MY (Fig 4B). We sought to disentangle these two QTL further to elucidate whether they are in pleiotropy or LD. However, due to high LD in the region and limitation in the data set (only 13 out of ~4,000 bulls carrying favourable recombinant haplotypes), this was not possible with the current data.

Additionally, we fine-mapped other dairy cattle traits, in an attempt to identify other potential selection target trait(s). Our results showed that the GC CNV was the lead variant for body condition score (BCS) and calving interval (CI; $-\log_{10}P$ = 21.4 and 6.6, respectively). Remarkably, GC CNV had a stronger association signal for BCS than it did for MY (S4 Fig). The CNs 4–6 allele, which was associated with low CM resistance, was correlated with low BCS (meaning low body fat content) and longer CI (meaning low fertility). The SNP association p-values obtained from either CM resistance and BCS or CM resistance and CI, clearly colocalized, underscoring that these QTL are driven by the same variant, which is likely to exert pleiotropic effects on each of these traits (S3 Fig and S4 Table).

## *GC* is the most functionally relevant gene underlying the CM resistance QTL on BTA 6

Our fine-mapping results hinted at transcriptional regulation as an underlying mechanism(s) of the CM resistance QTL, as our lead SNP was found in non-coding region. Thus, we mapped *cis*-expression QTL (further referred to as eQTL), to (1) identify shared variant(s) that are driving both local association and eQTL signals, and (2) to corroborate the causality of the candidate gene *GC* in the major CM resistance QTL. Prior to eQTL mapping, we firstly determined the most biologically relevant tissue(s) for our investigation. In the human transcriptome database [35–40], *GC* is predominantly expressed in the liver, whereas breast tissue showed no expression (S5 Table). Also, previous dairy cattle transcriptome studies showed that *GC* is expressed in the liver, kidney, and cortex, but not in the mammary gland [11,13,41]. Therefore, we performed eQTL mapping within the *cis*-regulatory range (+/- 1Mb region from the GC CNV), using liver RNA-seq data and Bovine HD genotype of lactating HF cows (n = 175). Since GC CNV is not in the BovineHD array, the GC CNV genotypes were obtained by (1) imputing BovineHD genotypes to WGS level variants and (2) genotyping the GC CNV directly. Direct genotyping was done by targeting five polymorphic sites (CN 1 as reference allele and CNs 4–6 as alternative allele) in the GC CNV (S6 Table). The best probe (BTA6:88,683,517) among the five showed 100% compatibility with the imputed GC CNV genotypes, underlining that our imputation approach was robust. Hence, we used the imputed genotypes (BTA6:87–89 Mb) to map eQTL further. The RNA-seq data was mapped using a reference guided method, where both reference annotated transcripts and novel transcripts can be discovered. Our RNA-seq data detected two different *GC* transcript isoforms: a canonical and an alternative transcript (Fig 5A). The former consisted of 13 exons, not encompassing the GC CNV, and accounted for a majority of overall *GC* expression (~98%). The latter shared the first 12 exons with the canonical transcript, however, it used an alternative 3' exon, the 14th exon, which is located inside the GC CNV. Expression of the alternative transcript was relatively low (~2% of the total GC expression). The reference gene set used for transcript assembly included the canonical form, but not the alternative transcript. Thus, *GC* gene-level eQTL mapping was done on the canonical form, and we additionally mapped transcript-level eQTL for the alternative *GC* transcript. Of the 13 genes annotated within the *cis*-regulatory range of CNV (+/- 1 Mb), *GC* was abundantly expressed ($\geq$5,000 transcripts per million (TPM)) and other genes were either lowly expressed (0.1 < TPM < 50), or not expressed (Fig 5B). Our gene-level eQTL mapping results discovered highly significant *cis*-eQTL for *GC*. The GC *cis*-eQTL was found in the BTA 6:88.68–88.88 Mb region ($-\log_{10}P > 23$; Fig 5D). The GC CNV was one of the top variants within the eQTL peak ($-\log_{10}P = 24.4$) and was in high LD with the GC eQTL lead SNP ($-\log_{10}P = 25.4$; $r^2 = 0.88$). P-values for GC expression and CM resistance were highly correlated ($\rho = 0.68$; Fig 5E), where CNs 4–6 were associated with increased *GC* expression and low CM resistance (Fig 5F). Additionally, *cis*-eQTL for Solute Carrier Family 4 Member 4 gene (*SLC4A4*), involved in bicarbonate secretion and associated with renal tubular acidosis, was found in BTA 6:88.67–89.07 Mb ($-\log_{10}P > 7$, S5 Fig). The lead SNP for *SLC4A4* eQTL (BTA 6:88,672,979, $-\log_{10}P = 7.75$) was found ~9 kb away from GC CNV, which also showed high significance ($-\log_{10}P = 7.1$). The GC CNV and the *SLC4A4* eQTL lead SNP were in high LD ($r^2 = 0.99$). P-values obtained from *SLC4A4* eQTL mapping and CM resistance were highly correlated ($\rho = 0.82$), and CNs 4–6 were associated with increased *SLC4A4* expression (S5 Fig). Additionally, we mapped a transcript-level eQTL for the alternative *GC* transcript. Intriguingly, the eQTL signal was driven by GC CNV tagging SNPs, followed by the second most significant variant, GC CNV ($-\log_{10}P = 16.9$ and $16.4$, respectively; Fig 5G). P-values for alternative *GC* transcript expression and CM resistance were correlated even stronger

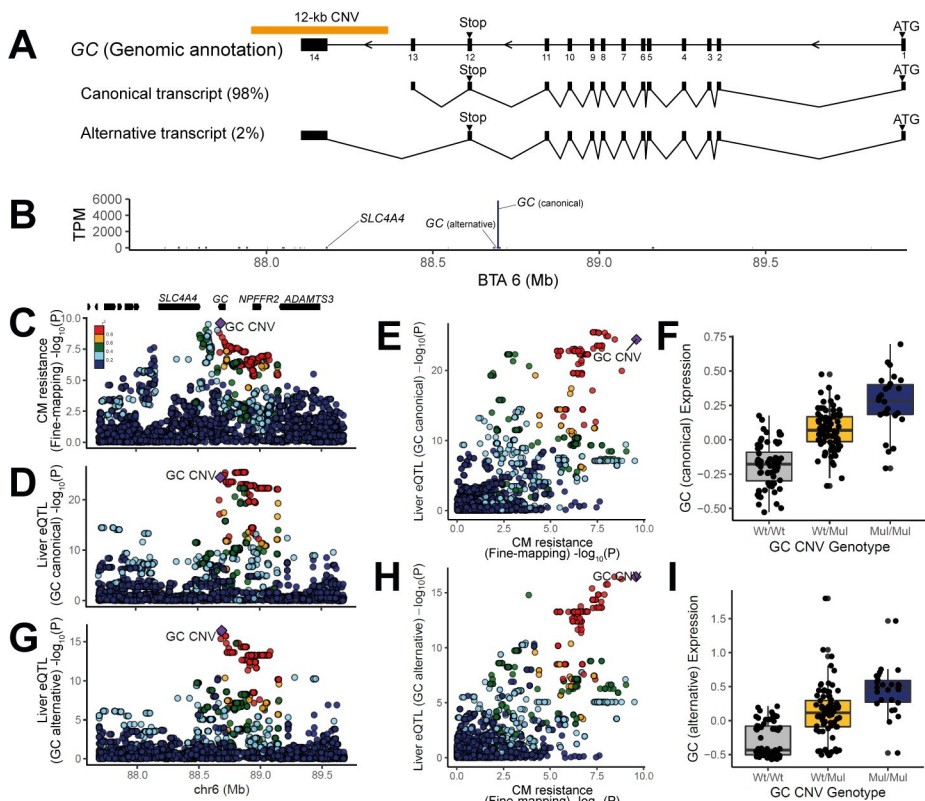

**Fig 5. eQTL mapping and colocalization of fine-mapping and eQTL mapping results for *GC* and the non-coding RNA.** (A) A Schematic overview of the *GC* gene structure and position of the GC CNV. Our data detected two *GC* transcripts, where the canonical form account the majority of the expression (98%) and an alternative form only counting for minor expression (2%). (B) eQTL was mapped for the genes located in a 2-Mb bin (BTA6:87.68–89.68). Of the 13 genes annotated in this bin, *GC* showed predominantly high expression (5,000 TPM <), whereas the rest were lowly expressed or not expressed at all. The eQTL were mapped for *GC* and *SLC4A4*. (C) CM resistance fine-mapping results were shown for the 2-Mb bin, where eQTL was mapped. The color scale indicates the degree of pair-wise LD ($r^2$) between the GC CNV and other SNPs. Annotation of genes in this region is drawn as black bars. Six genes on the left part are *AMBN*, *JCHAIN*, *RUFY3*, *GRSF1*, *MOB1B*, and *DCK*. (D) eQTL mapping results for *GC* (canonical transcript). (E) P-values obtained from CM resistance fine-mapping and GC eQTL mapping (canonical transcript) were correlated. The GC CNV is located in the right upper corner ($\rho$ = 0.68), showing that it is significant for both fine-mapping and eQTL mapping. (F) The box plot shows altered *GC* (canonical transcript) expression depending on GC CNV genotypes. (G) eQTL mapping result for GC (alternative transcript). (H) P-values obtained from CM resistance GWAS and GC eQTL mapping (alternative transcript) were correlated. The GC CNV is located in the right upper corner ($\rho$ = 0.74), showing that it is significant for both fine-mapping and eQTL mapping. (I) The box plot shows altered *GC* (alternative transcript) expression depending on GC CNV genotypes. Panels C-E, G, H were made with LocusCompare programme [99].

than the canonical transcript ($\rho$ = 0.74; Fig 5H), where CNs 4–6 corresponded to an increased expression of the alternative *GC* transcript (Fig 5I).

Our results indicate *GC* as a promising candidate gene, given the strong eQTL signal. On the contrary, high LD between GC CNV and the *SLC4A4* eQTL lead SNP ($r^2$ = 0.99) implied that *SLC4A4* could be a candidate gene. To prioritize between the two candidate genes, *GC* and *SLC4A4*, we used summary data-based Mendelian randomization (SMR) analysis [42], which estimates associations between phenotype and gene expression, aiming at identifying a functionally relevant gene, underlying GWAS hits. Our result showed *GC* to be the only gene whose expression was significantly associated with CM resistance association ($-\log_{10}P$ = 6), whereas *SLC4A4* was below the statistical threshold ($-\log_{10}P$ = 4.9; S7 Table). A subsequent analysis, heterogeneity in dependent instruments (HEIDI), was conducted to test whether the

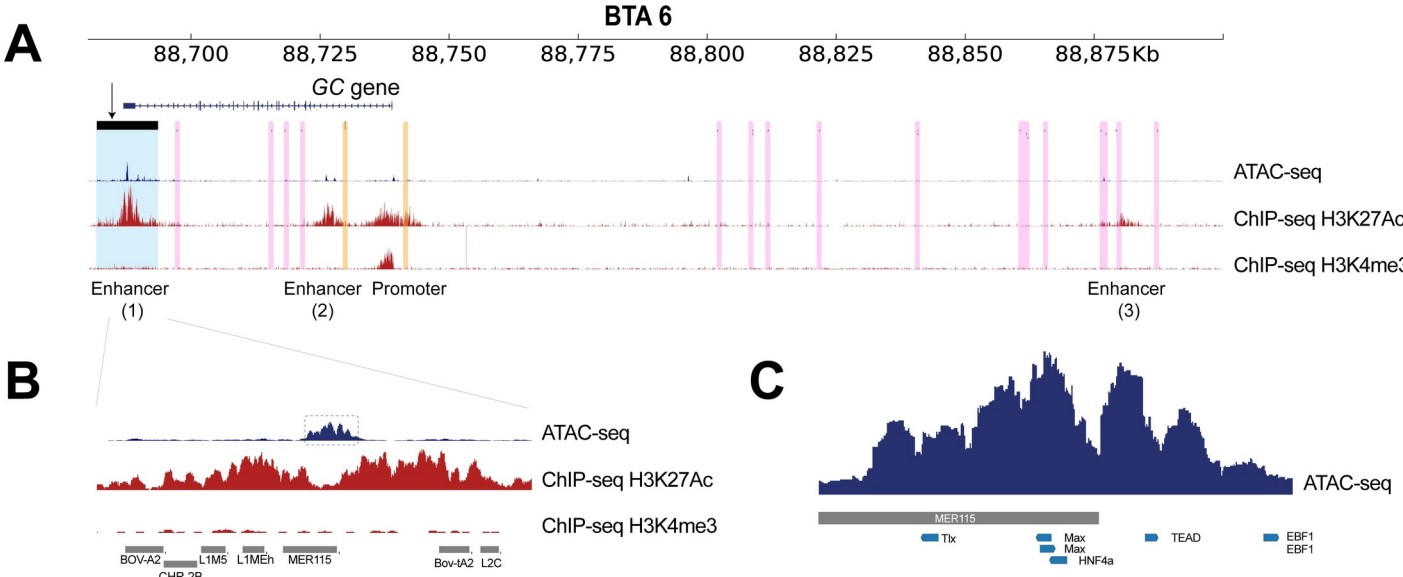

**Fig 6. Inspection of functional elements near the GC CNV.** Functional elements were inspected in *GC* eQTL region using ChIP-seq (H3K27ac and H3K4me3) and ATAC-seq data. The GC CNV genotype of the ATAC-seq sample was Mul/Mul (inferred based on CNV tagging SNP genotypes and read-depth increase in WGS data). The GC CNV genotype of the ChIP-seq sample was unknown (S9 Table).(A) The *GC* eQTL region was zoomed in. In this region, *GC* is the only annotated gene. The GC CNV is marked with the translucent blue, and CM resistance candidate SNPs reported by other studies [13,14] are marked with translucent yellow. Other significant eQTL lead SNPs in this region are marked with translucent pink. We overlaid ChIP-seq data to identify putative enhancers and promoters (ChIP-seq tracks; red). Furthermore, liver ATAC-seq data revealed highly accessible chromatin regions, supporting the regulatory elements discovered by ChIP-seq data sets (ATAC-seq tracks; blue). (B) We further zoomed in to the ATAC peak within the GC CNV, and discovered that the ATAC peak overlaps with MER 115. The predicted hepatic transcription factor binding sites are marked with translucent grey. (C) Transcription factor binding motifs are shown together with the ATAC signal located inside the GC CNV.

significant association between association hit and eQTL shown for *GC* was induced by a single underlying variant or two variants that are in LD (i.e. association hit variant is in LD with eQTL lead SNP). The result suggested a single underlying variant modulating both CM resistance association and *GC* eQTL ($P_{HEIDI}$ = 0.06). Thus, we confirmed *GC* as the most promising causal gene underlying CM resistance QTL, whose expression affects the CM resistance phenotype.

## A putative enhancer located in the *GC* CNV likely modulates the level of *GC* expression

We subsequently exploited epigenomic data sets to infer the functions of candidate variants in the 200-kb *GC* eQTL region (BTA 6:88.68–88.88 Mb; Fig 6A). Bovine liver epigenomic data, interrogating two histone modifications (ChIP-seq for H3K27ac and H3K4me3 marks) [43] and open chromatin regions (ATAC-seq) were investigated to infer functional contexts (i.e. active promoters and enhancers). The ChIP-seq data predicted one active promoter and three active enhancers, overlapping with ATAC-seq peaks, supporting the active status of the regulatory elements along *GC* (Fig 6A). Of the three putative enhancers identified, *GC* harbours two putative enhancers, one in the 1st intron and the other inside the GC CNV. The one located inside the GC CNV (further referred to as GC CNV enhancer) could be considered highly active, given the strong H3K27 acetylation mark. According to the comparative genomics catalogue of regulatory elements in liver [43], the GC CNV enhancer is cattle-specific, whereas the intronic enhancer is conserved between human, mouse, cow, and dog. These two putative enhancers were both supported by corresponding ATAC-seq signals, with a stronger peak

shown at the GC CNV enhancer (Fig 6A). Additionally, this strong ATAC signal overlapped with a repetitive element, MER115, from the DNA transposon family, hAT-Tip100 (Fig 6B). Some DNA transposons obtain enhancer function during evolution [44], and hAT-Tip100 was shown to function as enhancer in humans [43]. However, although the human ortholo-gous region harbours MER115, this repeat did not show enhancer activity in humans [40], underlining cattle-specific enhancer activity. Finally, we scanned the ATAC peak region inside the GC CNV, searching for transcription factor binding (TFB) motifs using Homer [45]. We found strong evidence for five motifs, including transcriptional enhancer factor (*TEAD*) and hepatocyte nuclear factor 4 alpha (*HNF4A*), supporting for the presence of the active enhancer within the GC CNV (Fig 6C).

## Discussion

In this study, we dissected a prominent CM resistance QTL in Dutch HF cattle, by integrating fine-mapping, eQTL mapping, and functional (epi)genomics data. Our findings revealed that the lead variant is the GC CNV, a 12-kb multi-allelic CNV, which harbours a putative *cis*-regu-latory element which targets *GC*. This CNV drives both the CM resistance association and the *GC* eQTL signals, underscoring *GC* as the likely causal gene underlying this QTL.

Identifying causal variants from GWAS can be challenging, since, often non-causal SNPs in high LD with the causal variant appear as lead SNPs [30]. Notably, our lead candidate SNP (rs110813063) has not been considered a strong candidate in other fine-mapping studies [7,13,14]. The function of our lead SNP, located 4-kb downstream of *GC*, was equally elusive, compared to other candidate SNPs that were located in the intronic region of or upstream of *GC* [7,13–15]. Nonetheless, the allelic imbalance pattern in our candidate SNP (Fig 3C), together with a previous report about a CNV in high LD with the GWAS candidate SNP [13], motivated us to hypothesize that the CNV might be the causal variant. As expected, we corrob-orated that our lead SNP, rs110813063, is a perfect tag SNP of the GC CNV ($r^2 = 1$). There are several explanations why previous fine-mapping studies missed rs110813063. Possibly, this SNP was absent in the GWAS variant set, as the standard SNP quality control (QC) criteria (i.e. removing SNPs with high depth and/or unbalanced allelic ratio) tend to eliminate SNPs inside CNVs. Alternatively, rs110813063 was present, but wrongly genotyped, due to highly disproportional allelic depth (Fig 3C). Hence, one may wonder whether a SNP-based GWAS approach, relying on a stringent QC, is sufficient in identifying causal variants, in a form other than point mutations. The answer might depend on the type of variant: studies showed that most deletions are well captured by tagging SNPs, whereas most duplications and multi-allelic CNVs are poorly tagged [46,47]. Note that the EHH analysis could clearly discern the GC CNV carrying haplotype (Fig 4B), showing the merit of a haplotype-based approach in delin-eating trait-associated structural variations, as demonstrated in earlier examples [48–51]. Also, an exploratory check for presence of CNVs might be beneficial for fine-mapping studies.

To connect the discovery from statistical associations to the molecular function, we showed that the association mapping signal was driven by the underlying molecular signal, expression of *GC* (Fig 5). Many heritable diseases are manifested in a tissue-specific manner [52]. Our trait of interest, CM, is manifested in the mammary glands, and hence it was considered a bio-logically relevant tissue for eQTL mapping. However, both large-scale human transcriptome databases (S5 Table) and cattle transcriptome studies indicated liver as the main organ of *GC* expression [11,13]. This discrepancy shows that prior knowledge of tissue-specific manifesta-tion of a trait is crucial for elucidating its molecular basis. In cattle, eQTL data was generated from diverse tissues (adrenal gland, blood, liver, mammary gland, milk, and muscle) [53–59], and some studies utilized the data sets in confirming causality of candidate genes [53–55,57].

We expect that the availability of eQTL data sets of diverse tissues and cell types will lead to rapid discovery and confirmation of candidate genes in farm animals in the future.

Bovine epigenomic data sets were utilized to prioritize candidate variants for the CM resistance QTL and eQTL for *GC* expression. The ChIP-seq and ATAC-seq data supported three active enhancers and one active promoter in the region of interest (Fig 6A). Of these regulatory elements, the GC CNV enhancer is considered the most likely causal variant, as multiple copies of enhancers can increase the target gene expression [60]. Hence, we propose that altered *GC* expression, mediated via multiplicated enhancers, is likely the key regulatory mechanism underpinning this QTL. Interestingly, candidate causal variants reported by previous studies [13,14] were found in the 1st intron of or upstream of *GC*, where an active enhancer and an active promoter were found (Fig 6A). In humans, tissue-specific enhancers outnumber protein coding genes [40,61]. Hence, a gene can be regulated by more than one enhancer, and a secondary enhancer is referred to as a shadow enhancer [62]. A recent study showed that redundant enhancers function in an additive manner, thus conferring phenotypic robustness (e.g. activities of multiple enhancers act together, thus removal of an enhancer still results in discernible phenotypes) [63]. In light of this finding, we speculate that the two enhancers located in *GC*, might have additive effects on *GC* expression.

Mammalian enhancers evolve rapidly, compared to promoters [43]. Also, rapidly evolving enhancers were exapted from ancestral DNA sequences and associated with positively selected genes [43]. The GC CNV enhancer is likely one of these rapidly evolving enhancers, given that (1) it exapted MER115, which does not function as enhancer in other species, (2) it is found in a selective sweep which harbors *GC*, and (3) it is cattle-specific. These findings strongly suggest that utilizing epigenomic data of species other than the one of interest, can be misleading, as it lacks species-specific regulatory elements. This provides a compelling reason to build species-specific epigenome maps, as already embarked upon by the international Functional Annotation of Animal Genomes (FAANG) project [64,65]. With this community effort, a wider range of species-specific regulatory elements underlying economically important QTL is expected to be unraveled in the future.

The major CM resistance QTL is known to have antagonistic effects for MY and our results confirmed this [13,14]. The trade-off between CM resistance and MY suggests that this locus might be under balancing selection [66]. One of the drivers of balancing selection is strong directional selection, which is common in livestock breeding [67]. Of the two traits of interest, MY has been the primary goal of dairy cattle breeding [34], and hence it seems plausible to assume that this locus is under balancing selection. Next to this, we had a particular interest in understanding the genetic basis of the antagonistic effects. The genetic modes considered were (1) a single pleiotropic variant affecting two traits and (2) two independent causal variants for each trait, in high LD. In case of pleiotropy, a particular genomic region cannot enhance both traits simultaneously through breeding. On the contrary, in case of LD, selection for recombinant haplotypes, containing favourable alleles for both traits, enables simultaneous improvement on the two traits. Only a small number (0.3%) of the studied animals had recombinant haplotypes (13 out of 4,142 bulls), and hence our data set lacks sufficient power to distinguish LD from pleiotropy. Future studies might consider two approaches for delineating this issue. The first is to exploit a daughter design by obtaining sufficient daughters of the 13 recombinant haplotype carrier Dutch HF bulls [68]. Another possibility would be to harness data from different breed(s). A dairy cattle breed that has both MY and CM resistance recorded, yet with low LD in the QTL region, would be most useful. Otherwise, a meta-GWAS of multiple cattle populations can aid in distinguishing between LD and pleiotropy.

We attempted to integrate the findings from the current study and further speculate on how *GC* expression and CM resistance are linked, assuming that the level of *GC* expression

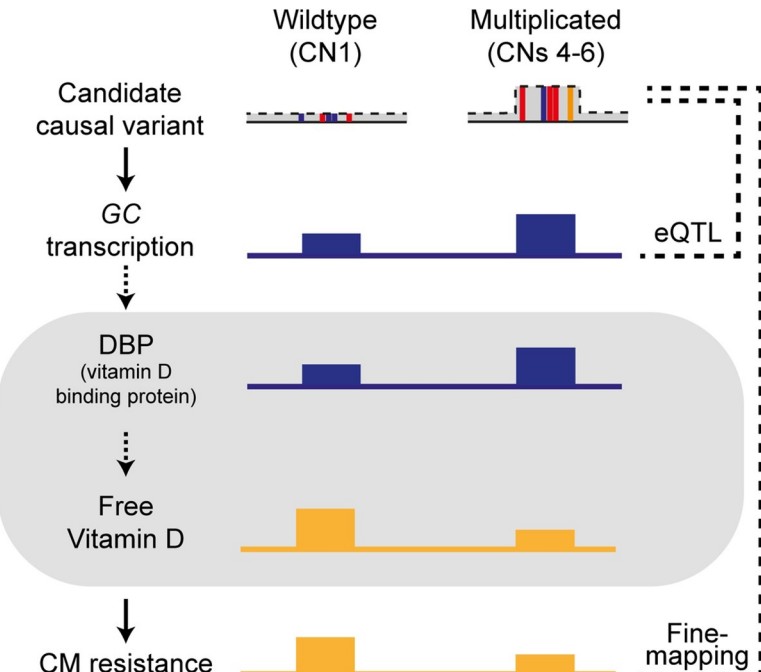

**Fig 7. Summary of the key findings and hypothesis of physiological aspects linking *GC* expression and CM resistance.** A schematic overview summarizing the allele effects of wildtype (CN 1) and multiplicated (CNs 4–6) alleles of the GC CNV, a likely causal variant for the major CM resistance QTL. The two alleles at the GC CNV locus lead to altered *GC* transcription, where the multiplicated alleles correspond to high *GC* expression. On the bottom shows the phenotypic association between the GC CNV and CM resistance, where the multiplicated allele is associated with low CM resistance. Finally, the area marked with grey shade shows our hypotheses that the amount of DBP is positively related with the GC expression. Further, we speculated that the amount of DBP and free vitamin D is inversely correlated, as long as vitamin D is bound by DBP, it is not biologically available. The solid arrows indicate the relations based on our findings. The dotted arrows indicate the relations based on our speculation.

and DBP is correlated; meaning absence of post-transcriptional /translational regulation (Fig 7). DBP binds to and transports vitamin D [69], yet it plays additional roles in bone development, fatty acid transport, actin scavenging, and modulating inflammatory responses (see [23,70] for review). To start with, DBP modulates immunity as a macrophage activating factor (DBP-MAF), when deglycosylated via glycosidases of T- and B- cells [71,72] and therefore enhances the immune response. Accordingly, we could hypothesize that CNV carriers have larger amounts of DBP, and subsequently improved immunity, leading to higher CM resistance. However, this hypothesis contradicts with our findings, as CNV carriers were shown to have lower CM resistance. Alternatively, DBP regulates the amount of freely circulating vitamin D metabolites [73]. According to the free hormone hypothesis, only free vitamin D metabolites are able to cross the cell membrane, and are thus biologically available [74]. In humans, only 0.03% of 25(OH)D, an indicator of vitamin D, is free, whereas the majority of vitamin D is bound either to DBP (85%) or to albumin (15%) [73]. Under these circumstances, CNV carriers, having a large pool of DBP, can be hypothesized to have lower levels of biologically available vitamin D, as postulated in human studies [75,76]. Thus, presumably, if CNV carriers have low amounts of free vitamin D, which may be termed as 'vitamin D deficiency', low CM resistance in these animals seems like a logical consequence (Fig 7).

Although vitamin D has been considered crucial in bone health [24], a recent review showed an inverse correlation between vitamin D concentrations and an extensive range of ill-health outcomes in humans [77]. Given the pervasive associations between vitamin D and

health conditions, there may be other associated traits induced by vitamin D deficiency. Indeed, GC CNV showed strong associations with BCS and CI, which are known to be negatively correlated with MY (e.g. cows with low BCS have longer CI and higher MY [78,79]), indicating pleiotropy (S4 Fig). The CNs 4–6, which were associated with low CM resistance, were associated with longer CI, meaning poor fertility. CI, a measure of duration from one calving to the next, is an indicator of fertility issues such as perturbations in the oestrous cycle, and/or anovulation [80]. Human studies reported ovarian disfunction induced by vitamin D deficiency [81]. This finding provides convincing evidence that the GC CNV might be the underlying variant inducing vitamin D deficiency and suboptimal female fertility. Furthermore, pleiotropic effects of the GC CNV indicated that CNs 4–6 were associated with low CM resistance and low BCS. This finding fits with the results found in HF cattle where low BCS was genetically correlated with high disease incidence [82], although the causal relationship between CM resistance and BCS remains unknown. Intriguingly, human studies showed contradictory results: obesity, a condition analogous to high BCS, predisposes patients to vitamin D deficiency, leading to disease susceptibility [51]. These opposing consequences of body composition traits possibly hint at an 'optimum' body fat amount, where an organism can function well without compromising its health.

## Conclusions

In this study, we dissected the major CM resistance QTL on BTA 6, integrating fine-mapping, CNV calling, eQTL mapping, and functional prioritization of candidate variants. We revealed a multi-allelic CNV harbouring a strong enhancer targeting *GC*, as the likely causative variant. Our findings revealed that the candidate causal gene *GC* is likely regulated by an enhancer located in the GC CNV. We speculate that GC CNV carriers which were shown to have high *GC* expression would have a larger amount of DBP, and by extension, low amount of biologically available vitamin D. This physiological condition probably puts animals into a state comparable to vitamin D deficiency and leads to low CM resistance. Moreover, we report evidence of pleiotropic effects of the GC CNV for other economically important traits as BCS, CI, and MY, which likely revolves around the vitamin D pathway. The current study provides a novel example of multiplicated *cis*-regulatory elements playing pleiotropic roles on various polygenic traits in dairy cattle.

## Materials and methods

### Ethics statement

RNA-seq data used for eQTL mapping was obtained from liver biopsy samples collected from ~14 day post-partum HF cows (n = 178). The procedures had local ethical approval and complied with the relevant national and EU legislation under the European Union Regulations 2012 (S.I. No. 543 of 2012). The institutes involved in samples collection and the respective local ethical approval information is as following: 1) University College Dublin, approved by University College Dublin Animal Research Ethics Committee (approval number: AE18982/ P046), 2) Agri-Food and Biosciences Institute, approved by Animal Welfare Ethical Review Body of Livestock Production Science Branch (approval number: PPL2754), 3) Aarhus University, approved by Danish Veterinary and Food administration Animal Experiments Inspectorate (approval number: 2014–15–0201–00282), 4) Walloon Agricultural Research Centre, approved by ethical commission of Liège University (approval number: 14–1617), and 5) Leibniz Institute for Farm Animal Biology, approved by the State Office for Agriculture, Fishery and Food safety of Mecklenburg-Western Pomerania (approval number: MV 7221.3–1.1-053/ 13).

## Whole genome sequencing and variant discovery

**Whole genome sequence data.** The genomes of 266 Dutch HF animals were sequenced. These 266 animals were closely related animals, where 240 were forming parents-offspring trios. The biological materials were either from sperm (males) or whole blood (females and males). Whole genome Illumina Nextera PCR free libraries were constructed (550bp insert size) following the protocols provided by the manufacturer. Illumina HiSeq 2000 instrument was used for sequencing, with a paired end protocol (2x100bp) by the GIGA Genomics platform (University of Liège). The data was aligned using BWA mem (version 0.7.9a-r786) [83] to the bovine reference genome UMD3.1, and converted into bam files using SAMtools 1.9 [84]. Subsequently, the bam files were sorted and PCR duplicates were marked with Sambamba (version 0.4.6) [85]. All samples had minimum mean sequencing depth of 15X and the mean coverage of the bam files was 26X.

**SNP calling and imputation panel construction.** Variant calling was done using GATK Haplotype caller in N+1 mode. We applied Variant Quality Score Recalibration (VQSR) at truth sensitivity filter level of 97.5 to remove spurious variants. Using the trusted SNP and indel data sets which are explained elsewhere [86], we observed that the VQSR step filtered out GC CNV tagging SNPs, possibly due to the overwhelmingly high depth in this region. Yet, GC CNV tagging SNPs were considered crucial in our research, as we considered that they could tag the GC CNV. Therefore, the genotypes of the GC CNV tagging SNPs obtained from the raw variant calling format (VCF) file were inserted in the VQSR filtered VCF file. While inspecting the CNV tagging SNPs, we discovered genotyping errors (three errors among 5 tagging SNPs of 266 individuals), where Ref/Alt was wrongly genotyped as Alt/Alt, due to severe allelic imbalance (where the number of alternative allele supporting reads is predominantly high). Using parent-offspring relationships available in our data set, we confirmed that these were true errors, and hence they were manually corrected. Finally, the VCF file of sequence level variants in BTA 6:84-94Mb, containing manually corrected GC CNV tagging SNPs was obtained. This VCF file, consisting of 45,820 variants of 266 animals was used as an imputation panel for fine-mapping analyses. For analyses related to haplotype segregation among the 266 sequenced animals, haplotype sharing among carriers and identification of selection signatures, we further applied stringent variant filters on the VCF file as explained in [87] to conserve only highly confident variants and to remove spurious genotyping and map errors.

## QTL mapping and fine-mapping analysis

**Phenotype and genotype data.** We obtained BTA6 genotype and phenotypic data of 4,142 progeny tested HF bulls from Dutch HF cattle breeding programme (CRV B.V., Arnhem, the Netherlands). The phenotype data was consisting of 60 traits, which are routinely collected in the breeding programme, including clinical mastitis resistance (S8 Table). CM resistance was recorded as a binary trait, depending on the disease status registration done by farmers and on somatic cell count level of routine milk recording samples [88]. The estimated breeding values (EBVs; obtained based on BLUP as published in the national evaluation by cooperation CRV (Arnhem, the Netherlands) [88]) were de-regressed to correct for the contribution of family members and de-regressed EBVs were used as phenotypes in an association analysis. The effective daughter contributions of the 4,142 bulls for CM resistance ranged between 25 and 971.3, with an average of 204.3 and a standard deviation of 217. The 4,142 bulls were genotyped with a low density genotyping array (16K). Afterwards, the 16K genotype data was imputed in two steps, firstly to 50K density, based on two private versions of a Bovine 50K genotyping array, and the panel was consisting of 1,964 HF animals. It was further imputed to a higher density, using a panel of 1,347 HF animals genotyped with Illumina

BovineHD BeadChip (770K). Subsequently, this genotype data was imputed to sequence level for the 10-Mb QTL region on BTA6 using the HF WGS imputation panel described above. The imputation was done with Beagle 4 [89], and variants with low minor allele frequency (MAF < 0.025) and low imputation accuracy (allele $R^2$ < 0.9) were filtered out.

**Association model and conditional analysis.** To confirm the presence of CM resistance QTL on BTA 6 in the Dutch HF population, we performed association mapping on BTA6 using imputed high density genotypes (28,669 SNPs on BTA6). The association mapping was performed SNP-by-SNP with a linear mixed model in GCTA [90]. The following model was fitted:

$$\mathbf{y} = \mathbf{1}\boldsymbol{\mu} + \mathbf{Xb} + \mathbf{g} + \mathbf{e}$$

where **y** is the vector of phenotypes (de-regressed EBVs), **1** is a vector of ones, **μ** is the overall mean, **X** is a vector of SNP genotypes coded as biallelic variant (0, 1, or 2), **b** is the additive effect of the SNP which is being tested for association, **g** is the polygenic effect captured by a genomic relationship matrix (GRM; random effect), and **e** is the residual. The GRM was built with GCTA, based on 50K genotypes to avoid losing statistical power by including causal markers [91]. The model assumed equal residual variances. A SNP was regarded significantly associated with CM, when -log$_{10}$P value was above 6.45 (chromosome-wide Bonferroni multiple-testing correction for 28,669 tests), at a nominal significance of p = 0.01. Subsequently, we repeated the association analysis in the BTA 6:84–94 Mb region, using the imputed sequence level variants (n = 45,820) to fine-map the QTL. The association model used was same as described above. Finally, to test if the lead SNP explained the QTL signal completely we ran a conditional association analysis for the imputed sequence variants in BTA 6:84–94 Mb region with the lead SNP as a covariate in the model.

## CNV discovery and characterization of GC CNV

The CNV was called from the WGS data of 266 animals, using the Smoove pipeline (https://github.com/brentp/smoove). This pipeline collects split and discordant reads using Samblaster [92], and then calls CNVs using Lumpy [93]. Afterwards, the CNV sites were genotyped using SVTyper (https://github.com/hall-lab/svtyper). Additionally, we used Duphold [94] to calculate read depth of the CNs at the GC CNV locus. Duphold exploits read depth between copy number variable regions and normal regions and calculates the ratio between these two. The integer diploid CNs obtained based on the Duphold coverage ratio values were assigned to the 266 animals. The CN distribution showed peaks at diploid CNs of 2, and 5–10, implying four haploid CNs (1, 4, 5, and 6) segregating at the GC CNV locus. Thus, possible haplotypic combinations for the diploid CNs would be: 2 (1/1), 5 (1/4), 6 (1/5), 7 (1/6), 8 (4/4), 9 (4/5), 10 (4/6 or 5/5). There was only one haplotypic combination possible for all the diploid CNs, except 10, were either (4/6) or (5/5) could form diploid CN 10. We confirmed the true presence of these CN alleles by taking advantage of our family structure. First, we verified that observed genotyped followed the Mendelian segregation rules. Next, we also checked that CN alleles transmission within the pedigree was in agreement with haplotype transmission. Haplotypes were reconstructed using familial information and linkage information using LINKPHASE3 programme [95]. The program estimates also, at each marker position and for each parent-offspring pair, the probability that a progeny inherited the paternal or the maternal haplotype for its parents.

To study the relationship between different CN alleles, we estimated homozygous-by-descent (HBD) probabilities at the CNV locus, and measured length of identified HBD segments. To that end, we ran with RZooRoH [96] a multiple HBD-class model described in [97],

with four HBD classes and one non-HBD class with rates equal to 10, 100, 1000, 10,000 and 10,000, respectively. As this approach compares haplotypes within individuals, it does not require haplotype reconstruction and is not affected by eventual phasing errors.

## Selection signature analyses

The BTA 6 was scanned for haplotype based selection signatures observed in the sequence variants from the 266 animals. We used the integrated haplotype homozygosity score (iHS [33]) using 'rehh' R package [32] for within-population analysis of recent selection signatures. In order to unravel the selection target trait(s) we identified a number of traits within the routinely collected catalogue of 60 traits showing QTL signals in the 84-94Mb region on BTA 6 based on 16K GWAS results (EuroGenomics custom SNP chip [98]). Hence, we performed GWAS for these traits (BCS, CI, and MY) in BTA 6:84-94Mb region based on imputed WGS variants to fine-map those QTL. The input files and the association model for the GWASs were the same as described above, except that phenotypes used de-regressed EBVs of BCS, CI, and MY, respectively. The GWAS results of these traits were plotted against the GWAS result of CM resistance to characterize the colocalization of QTL signals, using the R package "Locus-Comparer" [99].

## eQTL mapping and Summary data–based Mendelian randomization analysis

**RNA-seq data and eQTL mapping.** We used RNA-seq data produced by the GplusE consortium (http://www.gpluse.eu/; EBI ArrayExpress: E-MTAB-9348 and 9871) [100]. RNA-seq libraries were constructed using Illumina TruSeq Stranded Total RNA Library Prep Ribo-Zero Gold kit (Illumina, San Diego, CA) and sequenced on Illumina NextSeq 500 sequencer with 75-nucleotide single-end reads to reach average 32 million reads per sample. The reads were aligned to the bovine reference genome UMD3.1 and its corresponding gene coordinates from UCSC as a reference using HISAT2 [101]. Transcript assembly was conducted with StringTie [102], using reference-guided option for transcript assembly, which enables discovery of novel transcripts that are not present in the reference gene set. Reads were counted at gene- or transcript-level using StringTie. After data normalization using DESeq2 [103], we performed principal component (PC) analyses and removed outliers (PC > 3.5 standard deviations from the mean for the top four PCs, n = 2). Subsequently, the gene expression levels were corrected with Probabilistic Estimation of Expression Residuals (PEER) [104]. All animals were genotyped using Illumina BovineHD Genotyping BeadChip (770K). The genotype data was imputed to WGS level, using the imputation panel explained above, using Beagle 4 [89] and variants with low minor allele frequency (MAF < 0.025) and low imputation accuracy (allele $R^2 < 0.9$) were filtered out. Finally, the PEER corrected normalized gene expression was associated with the imputed WGS variants for 175 samples, using a linear model in R package "MatrixEQTL" [105].

**Prioritizing the causal gene.** We prioritized the most functionally relevant gene for the CM resistance QTL on BTA 6, using SMR (version 1.03) [42], to estimate association between phenotype and gene expression. Input data required were summary statistics from CM resistance fine-mapping and eQTL mapping results explained above. Additionally, the program requires plink format genotype data to analyse LD in the region of interest, for which the imputed WGS variants (BTA 6:84–94 Mb) of 4,142 bulls were used. A subsequent analysis, heterogeneity in dependent instruments (HEIDI), was conducted to test whether the significant association between lead hit and eQTL shown for GC was induced by a single underlying variant or two variants that are in LD (i.e. lead associated variant is in LD with eQTL lead

SNP). The statistical thresholds for SMR (-$\log_{10}$P>5) and HEIDI (P > 0.05) were benchmarked from the original paper [42].

**Functional genomics assay data.** The human transcriptome data bases were examined to find out in which tissue *GC* is highly expressed via Ensembl website (release 101; [106]). We downloaded liver ChIP-seq data (H3K27ac and H3K4me3) generated from four bulls from ArrayExpress (E-MTAB-2633 [43]). This ChIP-seq data was aligned to the bovine reference genome UMD3.1 using Bowtie2 [42] and peaks were called using MACS2 [107]. A catalogue of mammalian regulatory element conservation (https://www.ebi.ac.uk/research/flicek/publications/FOG15) was used to infer the conservation of the regulatory elements predicted from the ChIP-seq data sets. Next, ATAC-seq data was explored to see if chromatin accessible regions coincided with the histone marks obtained from the ChIP-seq data. We obtained a two weeks old male HF calf's liver ATAC-seq data from the GplusE consortium (ArrayExpress accession number: E-MTAB-9872). Data was analysed by following the ENCODE Kundaje lab ATAC-seq pipeline (https://www.encodeproject.org/pipelines/ENCPL792NWO/). Sequences were trimmed using Trimmomatic [41] and aligned on the bovine reference genome UMD3.1 using Bowtie2 [42]. After filtering out low quality, multiple mapped, mitochondrial, and duplicated reads using SAMtools [43] and the Picard Toolkit (http://broadinstitute.github.io/picard/), fragments with map length ≤146 bp were kept as nucleosome-free fraction. Genomic loci targeted by TDE1 were defined as 38-bp regions centered either 4 (plus strand reads) or 5-bp (negative strand reads) downstream of the read's 5'-end. ATAC-seq peaks were called using MACS2 [107] (narrowPeak with options—format BED,—nomodel,—keep-dup all,—qvalue 0.05,—shift -19,—extsize 38). We inspected the presence of an enhancer in the human orthologous region of the GC CNV, using ENCODE data [40] in the UCSC genome browser [108]. Transcription factor (TF) binding motifs in ATAC-seq peak regions were discovered using Homer [45]. The TF motifs that are expressed in bovine or human liver are kept and shown in the figure [109,110].

## EuroGenomics custom array genotyping

We attempted to directly genotype the GC CNV in a biallelic mode (CN 1 as reference allele and CNs 4–6 as alternative allele).

Probes targeting five polymorphic sites were designed in Illunima DesignStudio Custom Assay Design Tool were added to custom part of the EuroGenomics array [98] (S6 Table). DNA was extracted from the same biological material used for RNA-seq used for eQTL mapping (described above), and genotyped for the EuroGenomics array by the GIGA Genomics platform (University of Liège). The SNP genotypes were assessed using Illumina GenomeStudio software. Average call rate per probe was calculated to assess the quality of the probes. Finally, genotypes obtained from the highest quality (BTA6:88,683,517) was compared to the imputed GC CNV genotypes.

## Supporting information

**S1 Fig. IGV screen shots for the GC CNV and the breakpoints.** (A) The GC CNV (BTA 6:88,681,767–88,693,545) is shown in the IGV screen shot. The grey reads are normally mapped reads, whereas green ones are discordantly mapped reads, providing evidence for a tandem duplication. The sequencing coverage of the CNV region is higher than non-CNV region. (B) The left breakpoint is flanked by MIR repeat. (C) The right breakpoint does not overlap with repeats. (D, E) The proximal and distal breakpoints are zoomed in and the soft-clipped reads (positions where nucleotide sequences are written) information revealed the 5-bp microhomology "CACAT" at the breakpoints (marked as yellow). (F-H) A schematic

overview demonstrating a tandem configuration of the GC CNV. (F) The start and end of the GC CNV (shown in yellow-to-orange gradient colour) is flanked by non-CNV background, forming two junctions shown as 1–2 and 3–4 (marked with solid vertical lines). Sequencing reads from Wt (CN1) haploid genome can uniquely aligned to the 1–2 junction (blue-yellow reads) and the 3–4 junction (orange-green reads). The microhomology sequence is marked with grey shade. (G) A haploid CN4 genome forms unique junctions spanning over 3–2 formed by tandemly aligned 12-kb segments (marked with dotted vertical lines). Thus, reads from haploid CN4 would have reads spanning over 3–2 junction (orange-yellow reads), in addition to reads aligned to junctions 1–2 and 3–4. (H) Alignment of a heterozygous genome (CN1/CN4) to the reference genome (Wt) is shown. Both CN1 and CN4 have reads spanning over 1–2 and 3–4 junctions that are present in the reference genomes; these reads can be uniquely mapped. Reads spanning over 3–2 junctions cannot be uniquely mapped to the reference genome; these reads will be discordantly mapped either at 1–2 or 3–4 junctions (grey lines). The 3–2 junctions reads can be aligned to the junction 1–2, however orange reads will appear as soft-clipped. Likewise, these reads can be aligned to the 3–4 junction. However, the 4 junction started with the "CACAT" sequence. Hence, the sequences after the microhomology will appear as soft-clipped (marked with dotted line and asterisk).
(TIF)

**S2 Fig. Family tree of animals having CN 5 and CN6 allele on GC CNV locus.** In the panel of 266 animals, we observed CN 5 and CN6 alleles are mostly segregating among a small number of related animals. To show that CN5 and CN6 alleles are truly segregating, transmission probabilities at each marker position were calculated with LINKPHASE3. Transmission probabilities were estimated for paternal haplotypes (1 and 0 indicated transmission of the paternal and maternal allele, respectively). (A) Largest family of CN5 carriers. Each circle indicates one individual and the number inside the circle indicates the GC CNV copy number genotype. The numbers above each individual stand for the transmission probability. Question marks mean individuals with no copy number information. Male animals are shown as squares, female animals are shown as circles, and animals with unknown gender are marked with diamonds. (B) Largest family of CN6 carriers. The legends are identical to panel (A).
(TIF)

**S3 Fig. Chromosome-wide scan of selection signatures using integrated Haplotype Score (iHS) on BTA6.** Chromosome-wide scan of integrated Haplotype Score revealed two iHS peaks with high significance (-$\log_{10}$P > 5), near BTA6:78 Mb and BTA6:89 Mb.
(TIF)

**S4 Fig. Three traits that showed strong association signals in CM resistance QTL region on BTA 6.** Colocalization of fine-mapping p-values of a pair of traits are showing that body condition score (BCS), calving interval (CI), and milk yield (MY) are having association signal near or at the CM resistance QTL region on BTA 6. (A) Colocalization between body condition score and CM resistance is shown in the left panel, together with the separate Manhattan plots for body condition score and CM resistance on the right. (B) Colocalization between calving interval and CM resistance. (C) Colocalization between milk yield and CM resistance. Panel layout of (B) and (C) is the same as panel (A). In each panel, the colours of dots indicate degree of LD ($r^2$) with GC CNV (colour scale shown in the left upper corner of panel A). The purple diamond marks GC CNV. The figure was made with LocusCompare programme [99].
(TIF)

**S5 Fig. eQTL mapping and colocalization of fine-mapping and *SLC4A4* eQTL mapping results.** A colocalization plot for CM resistance fine-mapping and *SLC4A4* eQTL mapping

results is shown on the left side. The right upper panel is the CM resistance fine-mapping results and the right lower panel is the *SLC4A4* eQTL mapping results. In between these two right panels are the genes located in this region. Six genes on the left part are *AMBN*, *JCHAIN*, *RUFY3*, *GRSF1*, *MOB1B*, and *DCK*. In each panel, the colours of dots indicate degree of LD ($r^2$) with GC CNV (colour scale shown in the left upper corner of the colocalization figure. The purple diamond marks GC CNV.
(TIF)

**S1 Table. Positions of important variants in ARS-UCD1.2.**
(XLSX)

**S2 Table. Summary statistics for fine-mapping of clinical mastitis on BTA6 based on BovineHD data.**
(XLSX)

**S3 Table. Homozygosity-by-descent analysis results.**
(XLSX)

**S4 Table. GC CNV allele effects for CM resistance, milk yield, body condition score, and calving interval.**
(XLSX)

**S5 Table. GC expression across various tissue types in human transcriptome databases.**
(XLSX)

**S6 Table. Custom genotyping array probe design and genotyping results.**
(XLSX)

**S7 Table. Summary-based Mendelian Randomization analysis results.**
(XLSX)

**S8 Table. List of traits routinely collected in a Dutch HF breeding organization.**
(XLSX)

**S9 Table. GC CNV tagging SNPs genotypes of functional genomics data samples.**
(XLSX)

**S10 Table. BovineHD genotypes (BTA6:87.68–89.68Mb) used for eQTL mapping.**
(XLSX)

## Acknowledgments

The Dutch HF whole genome sequence population data set was funded by the DAMONA ERC advanced grant to MG. CC and TD are senior research associates from the Fonds de la Recherche Scientifique–FNRS (F.R.S.-FNRS). The authors are grateful to Elias Kaiser for discussion on the Vitamin D pathway, Ole Madsen for discussion on functional genomics data, Martijn Derks for providing advice on CNV calling pipeline.

## Author Contributions

**Conceptualization:** Michel Georges, Aniek C. Bouwman, Carole Charlier.

**Data curation:** Haruko Takeda, Gabriel Costa Monteiro Moreira, Latifa Karim, Erik Mullaart, Wouter Coppieters, Ruth Appeltant.

**Formal analysis:** Young-Lim Lee, Haruko Takeda, Tom Druet.

**Funding acquisition:** Roel F. Veerkamp, Michel Georges.

**Investigation:** Young-Lim Lee, Haruko Takeda, Tom Druet.

**Methodology:** Haruko Takeda, Tom Druet, Aniek C. Bouwman.

**Resources:** Erik Mullaart.

**Supervision:** Roel F. Veerkamp, Martien A. M. Groenen, Michel Georges, Mirte Bosse, Tom Druet, Aniek C. Bouwman, Carole Charlier.

**Visualization:** Young-Lim Lee.

**Writing – original draft:** Young-Lim Lee.

**Writing – review & editing:** Young-Lim Lee, Haruko Takeda, Gabriel Costa Monteiro Moreira, Latifa Karim, Erik Mullaart, Wouter Coppieters, Ruth Appeltant, Roel F. Veerkamp, Martien A. M. Groenen, Michel Georges, Mirte Bosse, Tom Druet, Aniek C. Bouwman, Carole Charlier.

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
