## [Editor Report · Decision Letter 0]

23 Mar 2021

Dear Dr Lee,

Thank you very much for submitting your Research Article entitled 'A 12 kb multi-allelic copy number variation encompassing a GC gene enhancer is associated with mastitis resistance in dairy cattle' to PLOS Genetics.

Our apologies for the delay in replying to your query, and thank you for your continued interest in PLOS Genetics. The concerns and issues regarding data sharing have been discussed by 6 members of the editorial board including the editors-in-chief. We outline a route forward below, but we also want to comment briefly on some of the central issues.

The rationale for PLOS' policy regarding open sharing of data is twofold: (1) to ensure that results can be validated and replicated; and (2) to allow other scientists to use the data in the future, for work that may or may not be related to the original work, and may not have been conceived at the time the original work was published. PLOS does recognize that there may be restrictions on human data sharing, but those restrictions are motivated and must be justified by respect for individual autonomy and privacy, and not by commercial or academic concerns on the part of the investigators, or to protect potential future applications of the data. We understand and appreciate the value of academic-commercial collaborations in the domestic animal space, and look forward to developing a broader conversation with Dr. Georges and additional colleagues regarding how to continually evaluate and refine PLOS policy.

Regarding the current manuscript, there is a consensus that the major accomplishment and potential impact of the work relates to the identification of the structural variant that affects GC activity. The path proposed in your email is generally acceptable:

(1) removing the genome-wide GWAS plot from the supplementary material;

(2) providing summary statistics for all of BTA6 as part of supplementary material or in a public database;

(3) revising the data availability statement to indicate that individual-level data is available without preconditions (e.g. authorship, collaboration) for academic use via an MTA.

In your revised manuscript, it will be necessary to make textual changes to the Results section to make clear that genome-wide genotyping data are not being reported. Instead, we suggest you focus primarily on BTA6 and describe in the materials and methods section how the genotypes for BTA6 were obtained.

We therefore ask you to modify the manuscript according to the review recommendations. Your revisions should address the specific points made by each reviewer.

[LINK]

Yours sincerely,

Tosso Leeb

Associate Editor

PLOS Genetics

Hua Tang

Section Editor: Natural Variation

PLOS Genetics

---

## [Decision Letter · Decision Letter 1]

4 May 2021

Dear Dr Lee,

Thank you very much for submitting your Research Article entitled 'A 12 kb multi-allelic copy number variation encompassing a GC gene enhancer is associated with mastitis resistance in dairy cattle' to PLOS Genetics.

The manuscript was fully evaluated at the editorial level and by independent peer reviewers. The reviewers appreciated the attention to an important topic but identified some concerns that we ask you address in a revised manuscript

We therefore ask you to modify the manuscript according to the review recommendations. Your revisions should address the specific points made by each reviewer.

[LINK]

Yours sincerely,

Tosso Leeb

Associate Editor

PLOS Genetics

Hua Tang

Section Editor: Natural Variation

PLOS Genetics

Reviewer's Responses to Questions

**Comments to the Authors:**

Reviewer #1: In this manuscript, Lee an colleagues describe the identification of a 12 kb multiallelic CNV associated with resistance to clinical mastitis and propose a GC gene enhancer in this CNV as the putatative causative mechanism. The authors exploited various data sets (genomic, transcriptomic, epigenomic) and conducted a thorough analysis using state-of-the-art methodolgy. Using mendelian randomaization and HEIDI to disentangle the two candidate genes e.g. is very convincing. On the other hand the authors clearly state the limits of their data sets that do not allow to differentiate between LD and pleiotropy in terms of the effect on milk yield vs. mastitis.

My only criticism would be that the manuscript could be written a bit more concisely (e.g. the part describing the CNV alleles).

Minor issues:

- In line 111, you cannot use the term 'nomal', I think.

- Regarding the SLC4A4 gene: an eQTL for this genes was also found - what is known about the function of this gene? Any implications?

- Effects for CI and BCS were found. These traits would be genetically correlated with MY, wouldn't they?

Reviewer #2: I thought was a very nice, well written paper. Proving a variant is causative is difficult without extensive lab work e.g. editing the functional site, but the authors make a strong case for the importance of the CNV. I have no major concerns about anything in the manuscript but there were some things that would be good if could be addressed.

I was a bit surprised the authors hadn’t tried to determine the actual sequence of the CNV. For example, in theory the extra copies of the sequence may not even come from this location i.e. they could be insertions somewhere else in the genome but because the extra copies aren’t in the reference genome all their reads map back to this location. The mendelian inheritance would suggest they are at least in the same region but at present just demonstrate more reads at this location than expected. Although one possibility is to Sanger across the breakpoints, as the CNV is relatively common there seems a reasonable chance one of the other current genome assemblies (Brahman, Angus etc) may have long reads/optical mapping data corresponding to the CNV. Or even the Hereford given could have been a heterozygote. Hadn’t seemed to check updated Hereford assembly and data. Understanding the sequence of the CNVs is likely important to understand its effects.

Along these lines I was a bit confused by Figure 3A. If the CTCCT, single copy haplotype was the ancestral state as the authors suggest I’m not quite sure how changed to multiple copies of TCTTG haplotype with no intermediate states being observed? Any hypotheses? Normally to infer ancestral state would use alignments to other breeds/species. E.g. what is the state of this region in the Brahman genome i.e. indicus breeds.

The authors clump copy numbers 4-6 together and analyse the CNV as if biallelic. Obviously if could show expression levels/phenotypes depend on the precise number of copies would be stronger evidence that the CNV may be causal.

Could impute expression values into wider GWAS cohort using a TWAS approach taking into account other cis-eQTLs to help differentiate between the two potentially regulated genes. These genes are unlikely to be solely regulated by just variants at this CNV region, so if the downstream phenotype is truly mediated by the expression level of the causal gene a more comprehensive TWAS approach incorporating the full set of cis-regulatory variants would potentially be informative.

More minor points:

I would probably have performed the eQTL analysis excluding any RNA-seq reads mapping to the CNV region i.e. just using the read counts across the other exons. As otherwise the results could be confounded.

Should be “did not have any evidence of a functional role”

“Bovine liver epigenomic data, interrogating two histone modifications (ChIP-seq for H3K27ac and H3K4me3 marks) [43] and open chromatin regions (ATAC-seq) [manuscript in preparation]” Would be good to know more details even if full details coming in another manuscript e.g. breeds studied etc. Can you infer the CNV state of the samples the chromatin came from by looking at the presence of the alleles in the reads that differentiate the CNV states? If discordance between their CNV number and the reference is mapping the chromatin data to the CNV complicated by the fact it is a CNV?

“The GC CNV and the SLC4A4 eQTL lead SNP were in high LD with GC CNV (r2 =0.99)” do you just mean “The SLC4A4 eQTL lead SNP is in high LD with the GC CNV (r2 204 =0.99)” The GC CNV should after all have an r2 of 1 with itself!

“We found strong evidence for five motifs, including transcriptional enhancer factor (TEAD) and hepatocyte nuclear factor 4 alpha (HNF4A), supporting for the presence of the active enhancer within the GC CNV (Fig 6C).” any disrupted by variants? Will their numbers change depending on copy number

Reviewer #3: The manuscript presented by Lee and collaborators reports a CNV spanning exon 14 of the group-specific component gene (GC) associated with resistance to clinical mastitis in Dutch Holstein Frisian cattle. The study is extraordinarily rich and dense, and combined multiple data sources to infer that the mentioned CNV affects an enhancer leading to increased expression of the canonical GC transcript, potentially reducing circulating vitamin D in carrier cows. Given the importance of the vitamin D pathway to immunity, and its previously hypothesized role in clinical mastitis, the results seem robust and sound. Apart from disagreeing with the use of the term “Vitamin D deficiency” in the absence of a detailed clinical and laboratorial characterization, I have only minor revisions to suggest which are listed below.

1 – The authors report that all analyses were performed according to the UMDv3.1 bovine assembly. Since ARS-UCDv1.2 has become the standard in the cattle community, the authors should provide the equivalent coordinates on the newer assembly. Instead of redoing part of the work, I would recommend the authors to simply report a liftOver analysis of at least the most relevant coordinates within the manuscript (ex. QTL region, CNV region, putative causal variant, and gene location). This could be done either within the text or as an additional tab in the existing supplementary file.

2 – Given the previous evidence from the literature supporting the QTL position in multiple populations, the GWAS analysis presented in this manuscript seems adequate and robust. However, for the sake of clarity, the authors should provide more information on the phenotypes used in their association mapping. More specifically, de-regressed estimated breeding values (dEBVs) were used as pseudo-phenotypes in the GWAS analysis, but no information of the models used to obtain these dEBVs was provided. For instance, a binary response variable was used for CM. Was the analysis based on a threshold liability regression model or a generalized linear mixed model (or something else)? Which fixed and random effects were included in the model? What was the number of records and the size of the numerator relationship matrix? How were cases and controls defined? Which methods were used to fit the model and under which settings (ex. AI-REML preceded by a number of EM iterations or Gibbs sampling with x burn in, y thinning interval and z samples)? How was the distribution of the resulting reliabilities of dEBVs (at least mean, sd and some percentile, say p90)? How were the reliabilities of EBVs computed? Since reporting this for all 60 traits may be difficult and tedious, I would encourage to do at least CM, given it is the main trait in the article. These data are important for future reference (e.g., replication of the results and comparative analyses in additional populations), and to understand how reliable were the pseudo-phenotypes since no weights were used for the residuals (recommended in the analysis of dEBVs). Perhaps the authors could include one more tab in the supplementary spreadsheet with this information, or a brief paragraph in the material and methods. If these data can be found somewhere else, please include a clear reference.

3 – In tab 1 of the supplementary file, please indicate the direction of the effect (which allele is being counted, A1 or A2?) and which allele frequency is reported (A1 or A2?). It seems a standard GCTA output, but it is better to confirm.

4 – In the IGV screenshots of Figure S1 I miss some description regarding the unmatching sequences observed in reads located in the flanks of the CNV (chimeric reads?). They indicate that the structural arrangement may not be perfect tandem duplications, and small sequence gains/losses likely occurred between successive copies (perhaps in a heterogenous way between CN4, CN5 and CN6). A quick alignment of these sequences within the segment could provide some insight, especially on the potential role of the MIR and CACAT microhomology as “guides” or anchor/pivot points to the duplications.

5 – In the discussion section the authors argue that SNP-based GWAS may miss relevant signals linked to structural variants, since these variants might be poorly tagged by SNPs because of their bi-allelic nature and of the possibility of genotyping errors caused by allelic imbalance within complex structural arrangements. They further suggest that exploratory analyses for the presence of CNV may be beneficial, especially with whole-genome sequence data. Wouldn’t it also be relevant to include a small discussion (1-2 phrases) on the possibility of using phased haplotypes as opposed to single markers to improve tagging? The EHH results presented here clearly suggest that haplotype data may be also beneficial (even with SNP Chip sparsity given the extended haplotype), and perhaps allow for prediction of causal variant carrier status in the discovery data set. This could be helpful as an additional tool to narrow down the candidate variants.

6 – I would include the following reference: Lippolis et al. (2011) 10.1371/journal.pone.0025479. The authors found that infusion of 25-Hydroxyvitamin D3 into the mammary gland of Holstein cows reduced the severity of experimental infection by Streptococcus uberis. Sample size was small (n = 10) but it is worth mentioning to further strengthen the authors’ hypothesis on the importance of the CNV in the etiopathogenesis of clinical mastitis in Holstein cows.

**Have all data underlying the figures and results presented in the manuscript been provided?**

Reviewer #1: Yes

Reviewer #2: Yes

Reviewer #3: Yes

PLOS authors have the option to publish the peer review history of their article (what does this mean?). If published, this will include your full peer review and any attached files.

Reviewer #1: No

Reviewer #2: No

Reviewer #3: No

---

## [Editor Report · Decision Letter 2]

8 Jun 2021

Dear Dr Lee,

We are pleased to inform you that your manuscript entitled "A 12 kb multi-allelic copy number variation encompassing a GC gene enhancer is associated with mastitis resistance in dairy cattle" has been editorially accepted for publication in PLOS Genetics. Congratulations!

Please make sure that all the accessions of underlying raw data are accurate and stated in the manuscript.

Yours sincerely,

Tosso Leeb

Associate Editor

PLOS Genetics

Hua Tang

Section Editor: Natural Variation

PLOS Genetics

Comments from the reviewers (if applicable):

**Data Deposition**

http://datadryad.org/submit?journalID=pgenetics&manu=PGENETICS-D-20-01910R2

**Press Queries**

---

## [Editor Report · Acceptance letter]

12 Jul 2021

PGENETICS-D-20-01910R2 

A 12 kb multi-allelic copy number variation encompassing a GC gene enhancer is associated with mastitis resistance in dairy cattle 

Dear Dr Lee, 

We are pleased to inform you that your manuscript entitled "A 12 kb multi-allelic copy number variation encompassing a GC gene enhancer is associated with mastitis resistance in dairy cattle" has been formally accepted for publication in PLOS Genetics! Your manuscript is now with our production department and you will be notified of the publication date in due course.

With kind regards,

Andrea Szabo

PLOS Genetics

On behalf of:
